# Rab3A/Rab27A System Silencing Ameliorates High Glucose-Induced Injury in Podocytes

**DOI:** 10.3390/biology12050690

**Published:** 2023-05-09

**Authors:** Olga Martinez-Arroyo, Ana Flores-Chova, Belen Sanchez-Garcia, Josep Redon, Raquel Cortes, Ana Ortega

**Affiliations:** 1Cardiometabolic and Renal Risk Research Group, Biomedical Research Institute of Hospital Clinico de Valencia INCLIVA, 46010 Valencia, Spain; omartinez@incliva.es (O.M.-A.); afloreschova@gmail.com (A.F.-C.); belen.snchezgarca@gmail.com (B.S.-G.); josep.redon@uv.es (J.R.); 2CIBEROBN (CIBER of Obesity and Nutrition Physiopathology), Institute of Health Carlos III, Minister of Health, 28029 Madrid, Spain; 3CIBERCV (CIBER of Cardiovascular Diseases), Institute of Health Carlos III, Minister of Health, 28029 Madrid, Spain

**Keywords:** diabetic nephropathy, gene silencing, podocyte, Rab3A/Rab27A system, vesicular traffic

## Abstract

**Simple Summary:**

In diabetic nephropathy, the injury of renal cells such as podocytes worsens the glomerular filtration process due to the cells’ loss and detachment from the basal membrane. Intra- and intercellular communications in cells are partially mediated by the Rab3A/Rab27A system. We have observed that silencing the Rab3A/Rab27A system in podocytes experiencing glucose overload has a differential impact on differentiation, cytoskeleton organisation, apoptosis, CD63 distribution, and miRNA levels. Generally, when cells are subjected to glucose stress and the Rab system is silenced, we observed a reduction in the detrimental cell processes. These results suggest that the Rab3A/Rab27A system is a key participant in podocyte injury and vesicular traffic regulation in diabetic nephropathy.

**Abstract:**

Diabetic nephropathy is a major complication in diabetic patients. Podocytes undergo loss and detachment from the basal membrane. Intra- and intercellular communication through exosomes are key processes for maintaining function, and the Rab3A/Rab27A system is an important counterpart. Previously, we observed significant changes in the Rab3A/Rab27A system in podocytes under glucose overload, demonstrating its important role in podocyte injury. We investigated the implication of silencing the Rab3A/Rab27A system in high glucose-treated podocytes and analysed the effect on differentiation, apoptosis, cytoskeletal organisation, vesicle distribution, and microRNA expression in cells and exosomes. For this, we subjected podocytes to high glucose and transfection through siRNAs, and we isolated extracellular vesicles and performed western blotting, transmission electron microscopy, RT-qPCR, immunofluorescence and flow cytometry assays. We found that silencing *RAB3A* and *RAB27A* generally leads to a decrease in podocyte differentiation and cytoskeleton organization and an increase in apoptosis. Moreover, CD63-positive vesicles experienced a pattern distribution change. Under high glucose, Rab3A/Rab27A silencing ameliorates some of these detrimental processes, suggesting a differential influence depending on the presence or absence of cellular stress. We also observed substantial expression changes in miRNAs that were relevant in diabetic nephropathy upon silencing and glucose treatment. Our findings highlight the Rab3A/Rab27A system as a key participant in podocyte injury and vesicular traffic regulation in diabetic nephropathy.

## 1. Introduction

Renal injury is a serious microvascular complication of diabetes mellitus, a highly prevalent disease currently reaching epidemic proportions [1]. Diabetic nephropathy (DN) occurs in up to 40% of diabetic patients, contributing to a worse quality of life and representing a substantial economic burden [2,3]. Physiologically, renal function is compromised, in part, by the weakening of the glomerular filtration barrier [4]. Podocytes are highly specialised cells that, in healthy kidneys, create an interdigitated network through foot processes surrounding the basal membrane to prevent plasma proteins from entering the urinary ultrafiltrate [5]. Permanent hyperglycaemia stimulates diverse signalling pathways and mechanical insults that act in an additive manner, thereby accelerating the podocyte damage [6,7]. Different molecular pathways lead to podocyte dysfunction through the inhibition of autophagy, accelerating epithelial to mesenchymal transition (EMT) or podocyte hypertrophy [8,9,10,11]. The Sirtuin 1 signalling pathway is closely related to damage in DN. Under increased glucose levels, the expression of Sirtuin 1 is inhibited, which increases oxidative stress and triggers apoptosis [12,13]. Moreover, in a hyperglycaemia environment, many factors activate the transforming growth factor (TGF-β1), which activates downstream pathways that ultimately stimulate the secretion of the vascular endothelial growth factor (VEGF), increasing the glomerular permeability [14].Furthermore, VEGF, growth hormone, reactive oxygen species, and Angiotensin II can stimulate TGF-β, leading to an accumulation of mesangial extracellular matrix, podocyte hypertrophy, mitotic effects and changes in its morphology [15,16,17,18]. When damaged, podocytes suffer foot process effacement, apoptosis, and detachment from the basal membrane, negatively affecting the filtration process [19,20].

Intracellular and intercellular communication between themselves and other renal cell types is imperative in podocyte function. In this scenario, extracellular vesicles (EVs) play a crucial role and become relevant partners in delivering “messages” through the wide variety of cargoes they transport, such as microRNAs (miRNAs), which regulate gene expression at different levels [21,22]. EVs are small vesicles classified into different types according to parameters such as size, content or cell/body fluid origin [23,24]. Rab GTPases (small GTPases such as the Rab3A/Rab27A system (Rab3A/27A system)) are an important family of proteins implicated in EV secretion. They have been demonstrated to be crucial for small EV (exosome) release, in charge of docking and fusion of multivesicular bodies with the plasma membrane [25,26,27,28].

In a previous study, we found a significant role played by the Rab GTPases of the Rab3A/27A system in podocyte injury, as evidenced by substantial mRNA and protein alterations under glucose overload [29]. However, this role remains understudied in renal diseases [30,31,32,33]. Therefore, investigating the involvement of the Rab3A/27A system by modifying gene expression could shed more light on the role of these Rabs in podocyte injury and reveal some of the cellular mechanisms of damage or protection guided by them. In this context, we aimed to silence gene expression of the Rab3A/27A system in human immortalised podocytes stressed by high glucose to gain more in-depth insight into their effect on podocyte morphology, apoptosis and function and ascertain how this silencing could affect vesicular traffic and their cargos.

## 2. Materials and Methods

### 2.1. Podocyte Cell Culture

Cell line AB8/13 of conditionally immortalised human podocytes was kindly provided by Prof Moin Saleem of the Children’s Renal Unit and Academic Renal Unit, University of Bristol, Southmead Hospital, Bristol, UK [34]. Briefly, podocytes were cultured in the Roswell Park Memorial Institute (RPMI) medium 1640 with 10% exosome-depleted foetal bovine serum (FBS) (Bio west, Nuaillé, France), 1% insulin-transferrin-selenium (ITS) (Gibco, Thermo-Fisher Scientific, Waltham, MA, USA) and 1% penicillin/streptomycin (P/S) (Biowest, Nuaillé, France). Cells were propagated until 80% confluence at 33 °C and replated at 70% confluence for differentiation. Next, podocytes were differentiated at 37 °C for 10 days with lower FBS concentration (2%) [29]. All experiments were performed on differentiated podocytes.

### 2.2. Cell Transfection and Glucose Treatment

The podocytes were subjected to transient transfection with silencing RNAs (siRNAs) to inactivate *RAB3A* and *RAB27A* genes, using Lipofectamine 3000 (Invitrogen, Thermo Fisher Scientific, Waltham, MA, USA) following the manufacturer’s instructions. Pre-designed siRNAs were utilised (Ambion™ Silencer™ Select Pre-designed siRNAs (Ambion™ Silencer™ Select Pre-designed siRNA) for *RAB27A* gene (ID: s11693) and a pool of two siRNAs for *RAB3A* (ID: s11667 and s11668) were employed. To control the correct transfection efficiency, we used Cy3™ Dye-Labeled Negative Control (Invitrogen, Thermo Fisher Scientific, Waltham, MA, USA). As a negative control or scramble (scr), we used Silencer™ Select Negative Control siRNA (Invitrogen, Thermo Fisher Scientific, Waltham, MA, USA), and as a positive control we used a siRNA from the constitutively expressed gene *GAPDH* (Silencer™ Select GAPDH siRNA, 4404024, Invitrogen, Thermo Fisher Scientific, Waltham, MA, USA) (Appendix A). After 24 h, the transfection medium was replaced by RPMI at 1% P/S and ITS. Subsequently, D-(+)-Glucose (Sigma Aldrich, St. Louis, MO, USA) was added at normal (NG; 5.5 mM) and high (HG; 30 mM) concentrations, and the cells were incubated for 24 h at 37 °C.

### 2.3. Exosome Isolation from Podocyte Cultures

Exosomes were isolated from the cell culture medium utilising a protocol based on sequential ultracentrifugation [35]. Briefly, the medium was collected, and cells and debris were eliminated by centrifugation for 30 min at 2250× *g*. Afterwards, cell-free supernatant was centrifuged for 45 min at 20,000× *g* (Ultracentrifuge Optima L 100K, 70 Ti rotor, Beckman Coulter, Indianapolis, IN, USA) to precipitate larger EVs. Next, the supernatant was centrifuged for 70 min at 110,000× *g* to obtain the exosome-enriched pellet, which was finally resuspended in 0.2 µm filtered phosphate-buffered saline (PBS) and processed for downstream analyses.

### 2.4. Nanoparticle Tracking Analyses (NTA)

To characteriseise EV populations, nanoparticle tracking analysis (NTA) was performed on a NanoSight LM10 (Malvern Panalytical Ltd., Malvern, UK). Samples were diluted 1/50 with filtered PBS to reach the concentration recommended by the manufacturer (20–120 particles/frame) and injected into the NanoSight, with a 405 nm laser and an sCMOS camera. Data were analysed by NTA software version 3.3 (Dev Build 3.3.104), with the Min Track Length, Max Jump Distance, and Blur parameters set to auto, and the detection threshold set to 5. The camera level was set to 15, and five readings of 30 s at 30 frames per second were taken with manual temperature monitoring.

### 2.5. Transmission Electron Microscopy (TEM)

To complete EV characterization, samples were observed by transmission electron microscopy (TEM). For immunogold labelling, 8 μL of EV samples were first fixed in 2% paraformaldehyde-0.1 M PBS for 30 min on carbon-coated nickel grids. Afterwards, the grids with adherent exosomes were washed in 0.1 M PBS and blocked in 0.1 M glycine and 0.3% bovine serum albumin (BSA) for 10 min. The grids were incubated with the primary antibodies CD9 and CD63 (rabbit and mouse, respectively) for 1 h. Following an additional 20-min blocking step, the grids were incubated for 1 h in appropriate secondary antibodies with 6 nm and 15 nm gold particles conjugated, respectively. Finally, after washing, a standard negative staining procedure was performed and observed under a transmission electron microscope (FEI Tecnai G2 Spirit, ThermoFisher Scientific Company, Eugene, OR, USA). All images were acquired using Radius software (Version 2.1) with a Xarosa digital camera (EMSIS GmbH, Münster, Germany).

### 2.6. RNA Extraction and cDNA Synthesis

The total RNA from the cell pellet was extracted with the miRNeasy Mini Kit (Qiagen, Hilden, Germany) following the manufacturer’s instructions. For RNA isolation from the exosome samples, we used the Total Exosome RNA and Protein Isolation kit (Invitrogen, Thermo Fisher Scientific, Waltham, MA, USA). The RNA concentration and the purity of the samples were measured by a NanoDrop 2000 spectrophotometer (Thermo Fisher Scientific, Waltham, MA, USA), after which the samples were stored at −80 °C until use.

For the mRNA analyses, cDNA was synthetised using the Ready-To-Go You-Prime First-Strand Beads kit (GE Healthcare, Buckinghamshire, UK) following the protocol indications. For the miRNA analysis, 2 μL of total RNA were used in the TaqMan™ Advanced miRNA cDNA Synthesis Kit (Applied Biosystems, Foster City, CA, USA) following the manufacturer’s instructions. The cDNA obtained for the two techniques was stored at −20 °C until use.

### 2.7. Quantitative Real-Time Polymerase Chain Reaction (RT-qPCR)

The Quantitative Real-Time Polymerase Chain Reaction (RT-qPCR) was assessed in the LightCycler 480 II real-time PCR system (Roche, Mannheim, Germany). Qiagen Multiplex PCR Master Mix with LC Green reagent (Qiagen, Hilden, Germany) and specific primers designed by Primer 3 software (Version 0.4.0) were used for mRNA analyses (Table 1). Additionally, 2 μL of diluted cDNA were combined with TaqMan^®^ Fast Advanced Master Mix (2×), and specific TaqMan™ Advanced microRNA assay probes (Applied Biosystems, Foster City, CA, USA) for miR-26a-5p (ID: 477995), miR-21-5p (ID: 477975) and miR-31-5p (ID: 478015) for the miRNA analyses. All assays were run in triplicate, including the appropriate controls and the non-template control. Furthermore, a melting curve analysis was performed to evaluate the specificity of the amplicons generated. Subsequently, β-Actin (*ACTB*) and β2-microglobulin (*B2MG*) housekeeping genes were used to normalise mRNA levels, and cel-miR-39-3p (ID: 478293) and ath-miR-159a (ID: 478411) were used as an external reference for the miRNA analyses. Finally, the relative amount for each target gene or miRNA was calculated by the 2^−ΔΔCt^ comparative method (where Ct is the threshold cycle). The values for each analysed gene and miRNA were expressed as the fold change between the target gene and the average value of the housekeeping genes.

### 2.8. Cell Pellet and Exosome Homogenization, Electrophoresis and Western Blot Analyses

We employed a radioimmunoprecipitation assay (RIPA) buffer (ThermoFisher Scientific, Waltham, MA, USA) with a protease inhibitor cocktail (Sigma-Aldrich, St. Louis, MO, USA) for podocyte cell pellet and exosome lysing. Next, supernatant was obtained by centrifugation at 15,000× *g* for 15 min at 4 °C. Protein quantification was assessed by the Lowry method, using BSA as the standard. Protein samples were loaded into NuPAGE 4–12% polyacrylamide gels or NuPAGE 3–8% tris-acetate gels (Invitrogen, Carlsbad, CA, USA) and were then transferred to polyvinylidene difluoride (PVDF) membranes. After membrane blocking, incubation with primary antibodies at room temperature (RT) was performed for 2 h. The primary antibodies used were: mouse monoclonal anti-Rab3A (1/100, Sigma-Aldrich, St. Louis, MO, USA); mouse monoclonal anti-Rab27A (1/500, Abcam, Cambridge, UK); rabbit polyclonal anti-Rabphilin3A (1/500, Sigma-Aldrich, St. Louis, MO, USA); rabbit polyclonal anti-synaptopodin (1/1000, Sigma-Aldrich, St. Louis, MO, USA); rabbit monoclonal anti-nephrin (1/500, Abcam, Cambridge, UK); rabbit polyclonal anti-Wilms tumour 1 (WT-1; 1/500, Abcam, Cambridge, UK), and rabbit polyclonal anti-CD2-associated protein (CD2AP; 1/500, Invitrogen, Thermo Fisher Scientific, Waltham, MA, USA). To characterise the EV fractions, we used an anti-tumour susceptibility gene 101 (TSG101, 1/500, Abcam, Cambridge, UK), anti-CD9 (1/1000, Abcam, Cambridge, UK), anti-CD63 (1/200, Abcam, Cambridge, UK), anti-syntenin (1/500, OriGene, Rockville, MD, USA), anti-calnexin (1/1000, Abcam, Cambridge, UK) and anti-GM-130 (1/1200, Abcam, Cambridge, UK). A mouse monoclonal anti-β-actin antibody (1/6000, Sigma-Aldrich, St. Louis, MO, USA) was used as the loading control. After washing with tris-buffered saline with Tween 20 (TBS-T, 20  mM tris-HCl, 150  mM NaCl and 0.1% Tween 20), the membranes were incubated with alkaline phosphatase-conjugated anti-rabbit IgG or anti-mouse IgG antibodies (Sigma-Aldrich, St. Louis, MO, USA) at RT for 1  h. Afterwards, they were washed three times with TBS-T and TBS, and chromogen 5-bromo-4-chloro-3-indolyl phosphate/nitro blue tetrazolium (BCIP/NBT, Sigma-Aldrich, St. Louis, MO, USA) was used to detect bound antibodies. Finally, the bands were digitalized and quantified by ImageQuant™ 7.0 TL software. All membrane blots of the whole gel are shown in Appendix A.

### 2.9. Immunocytochemistry

The immunofluorescence experiments were assessed following the protocol described in [13]. Briefly, after 4% paraformaldehyde fixation, permeabilisation and blocking with PBS + BSA (1%), the samples were incubated overnight at 4 °C with the same primary antibodies as for the western blot for the anti-CD63 (1/100), anti-CD9 (1/2000) in the PBS-Tween 20 (PBS-T) + BSA (1%) buffer. Next, the samples were incubated in darkness with an Alexa-conjugated secondary antibody (Invitrogen, Carlsbad, CA, USA) for 1 h at RT in the same buffer. Within this period, PBS-T + BSA (1%) with DAPI (1/1000) was added to identify the nuclei. After washing, the samples were mounted and observed using a DMi8 Leica confocal microscope (Leica Microsystems, Wetzlar, Germany) and images were processed with ImageJ (v. 1.46 r; National Institutes of Health, Bethesda, MD, USA) software. The distribution of the F-actin cytoskeleton was estimated by phalloidin staining on silenced and treated podocytes. For this purpose, the cells were fixed, washed, permeabilised, and incubated for 1 h at RT with phalloidin-iFluor 594 Reagent (1/1000, Abcam). DAPI was added within the last half hour. After incubation, the cells were washed, mounted, and observed. F-actin fibre orientation was measured with the FibriTool plugin [ImageJ] [36], which measures fibre disorder, quantifying the fractional anisotropy of fibrillary structures in raw images. Values close to 0 correlate to isotropic or disordered fibres, and values close to 1 indicate orientated filaments.

### 2.10. Cell Apoptosis Assay

The apoptosis of transfected and treated podocytes was measured by flow cytometry. After gene silencing and treatment with glucose, cell pellets were trypsinised, resuspended, centrifuged for 3 min at 1300 rpm, and washed. They were then resuspended in 200 µL annexin V binding buffer (Immunostep, Salamanca, Spain) and incubated in darkness for 15 min with 5 µL annexin V and 5 min with 7 µL 7-amino-actinomycin D (7-AAD). Finally, 400 µL of annexin binding buffer was added. A total of 10,000 events from each replicate were analysed in the flow cytometer (BD LSRFortessa X-20) with FACSDiva 8.0.1 software (BD Bioscience, Franklin Lakes, NJ, USA) [29].

### 2.11. Statistical Analysis

The data are expressed as fold change and mean ± standard error. The Kolmogorov--Smirnov test was used to analyse the normal distribution of the variables. Student’s *t* test or the Mann--Whitney U test were used for comparisons between two groups depending on data distribution. The level of significance was set at *p* < 0.05. The statistical analyses were performed with SPSS software (version 20, SPSS), and graphical plots were created with SigmaPlot (Version 10, Systat Software).

## 3. Results

### 3.1. Effect of RAB3A and RAB27A Gene Silencing and Treatment with High Glucose on the Rab3A/27A System and Podocyte Markers

To study any possible alterations in the Rab system resulting from the silencing of one of its components and the changes observed upon glucose overload, we analysed Rab27A expression when *RAB3A* was silenced and vice versa (siRab3A and siRab27A), as well as the levels of their common effector Rabphilin3A. Furthermore, to observe how Rab silencing or a combination of both silencing and glucose treatment induces alterations on podocyte structure and function, podocyte markers were studied.

Our results showed a decrease in the mRNA levels of each Rab when the other was silenced under NG conditions (−2.99 fold change (FC), *p* < 0.05 for siRab3A; −1.72 FC, *p* < 0.05 for siRab27A). However, the opposite effect was observed in the HG group, with increased expression of *RAB27A* when *RAB3A* is silenced and vice versa, although only the expression of siRab27A reached statistical significance (1.71 FC, *p* < 0.05) (Figure 1A,B). Protein levels were also analysed, but without reaching statistical significance. Rabphilin3A protein levels were decreased under NG conditions when Rab3A and Rab27A were silenced (−1.16 FC, *p* < 0.01; −1.15 FC, *p* < 0.05, respectively). In contrast, under HG conditions, Rabphilin3A levels were augmented under *RAB27A* silencing (1.14 FC; *p* < 0.001) (Figure 1C,D). These findings suggest that each Rab plays a collaborative role by increasing its expression levels during cellular stress to counteract the decrease of the other Rab.

To study the influence of Rab3A/27A system silencing and glucose overload on podocyte structural and functional proteins, our next step was to analyse the gene and protein levels of the podocyte markers synaptopodin, CD2AP, WT-1, and nephrin. In the case of *RAB3A* silencing, an overall mRNA decrease in all markers was observed in NG, as shown for synaptopodin (−3.93 FC, *p* < 0.01) and *CD2AP* (−2.18 FC, *p* < 0.05), whereas under glucose overload, an increasing trend was observed in all podocyte markers, significant for synaptopodin (3.15 FC, *p* < 0.01) and WT-1 (4.75 FC, *p* < 0.05) (Figure 2A). For protein levels, an increase in CD2AP and WT-1 was observed when *RAB3A* was silenced at HG concentrations (Figure 2B). In summary, these findings show that Rab3A silencing induces a general decrease in podocyte markers, while the opposite effect is observed with silencing under glucose overload.

In the case of *RAB27A* silencing, reduced synaptopodin (−3.12 FC, *p* < 0.05) and *CD2AP* (−2.32 FC, *p* < 0.05) mRNA levels were found in NG. Under HG conditions, a tendency to decrease was observed in synaptopodin levels, while unlike for NG, there was a marked increase in *CD2AP* (1.70 FC, *p* < 0.05) and nephrin levels (4.58 FC, *p* < 0.001) (Figure 3A). Regarding protein levels, a downward trend of synaptopodin and CD2AP levels was observed under NG conditions, contrasting with an increase in nephrin (1.32 FC, *p* < 0.001) in accordance with the changes at the mRNA level. In HG, *RAB27A* silencing produced a drop in synaptopodin (−1.19 FC, *p* < 0.05) and CD2AP (−1.22 FC, *p* < 0.01) and a rise in nephrin (1.45 FC, *p* < 0.05) (Figure 3B). Similar to what was observed with *RAB3A* silencing, *RAB27A* silencing also seems to induce dedifferentiation of podocyte when it is silenced without treatment, inducing the opposite effect under HG.

### 3.2. Influence of Rab3A/27A System Silencing and Glucose Overload on Podocyte Apoptosis and Structure

We next sought to determine the effect of both Rab3A/27A silencing and glucose treatment on apoptosis and the cellular structure of podocytes. First, flow cytometry assays showed that in both cases (*RAB3A* and *RAB27A* silencing), a significant increase in apoptosis was observed in the control (CNT) scr group compared with the CNT without- transfection group, evidencing an effect of lipofectamine on cell death (*p* < 0.05) (Figure 4 for *RAB3A* silencing and Figure 5 for *RAB27A* silencing). Similarly, increased cell death was noted in the groups treated with HG without silencing. Strikingly, *RAB3A* silencing caused a decrease in apoptosis at NG levels (*p* < 0.05, Figure 4), contrary to *RAB27A* silencing, which produced an increased apoptosis in an NG condition (Figure 5). However, in cells exposed to HG concentrations, apoptosis was lower in both Rab-silenced groups compared to the HG + (CNT scr) groups. This effect was more pronounced in the Rab3A-silenced group (*p* < 0.001, Figure 4B) than in the Rab27A-silenced group (*p* < 0.05, Figure 5B). This indicates that Rab gene silencing seems to cause a decrease in podocyte apoptosis when subjected to HG concentrations.

To assess the influence of Rab-system silencing and glucose overload on podocyte structure, we analysed F-actin fibre distribution by measuring the fractional anisotropy, which indicates how organised the F-actin fibres in the cell are. Similar results were found for both silenced Rabs, with a disorganised fibre network (anisotropy values close to 0) in HG-treated podocytes compared to the NG group (CNT), which was also observed in the CNT scr groups. Interestingly, under NG conditions, the silencing of both Rabs was associated with a decrease in anisotropy, whereas this was reversed when silencing both Rabs under HG (Figure 6).

Conjointly, these results provide further evidence of different cellular responses to detrimental processes, depending on whether gene silencing occurs under normal physiological conditions or in a diabetic environment.

### 3.3. Analysis of Vesicular Transport Markers under Rab3A/27A System Silencing and Glucose Treatment

Given the involvement of the Rab system in vesicular trafficking, we next studied the influence of *RAB3A* and *RAB27A* silencing and HG on the expression of CD63 and CD9, both common tetraspanins enriched in EVs. We performed fluorescence analyses of CD63 and CD9 and quantified the intensity of the images. In NG, silencing both Rabs resulted in an increased CD63 signal, with a change of distribution from a homogeneous and scattered dotted pattern throughout the cytoplasm to a pattern with more accumulation at the perinuclear zone. However, in HG and siRab, this was not as marked as in NG, and tended to resemble the CNT pattern, especially in the case of siRab27A (Figure 7). Conversely, the CD9 pattern and intensity did not show significant changes, but its intensity showed a tendency to decrease when Rabs were silenced under both NG and HG conditions. These observations could indicate that Rab3A/27A silencing may cause alterations in the vesicular transport CD63-associated pathway in podocytes.

### 3.4. Isolation and Characterization of Exosomes from Podocyte Culture Medium

To validate the EV isolation and purification protocol, vesicles were characterised by analysing their size distribution by NTA, morphology by TEM, and purity of EV suspension through the analysis of the vesicle marker expression by western blot. NTA analysis showed a marked peak of the selected EV fractions, ranging from 140 to 200 nm (Figure 8A). TEM images revealed pure samples without undesired precipitates or cell debris and showed vesicles stained with well-established EV markers CD63 and CD9 (Figure 8B). In addition, the western blot experiments confirmed the purity of the samples where vesicles were enriched in surface and cytoplasmic EV markers CD63, CD9, syntenin, and TSG-101. The intracellular compartment markers calnexin and GM-130 were highly enriched and exclusively expressed in the cellular pellet samples (Figure 8C). The use of various techniques provides evidence that the isolation and purification of EVs were executed accurately, resulting in an enriched EV pellet primarily composed of exosomes.

### 3.5. Effect of Rab3A/27A System Silencing on miRNA Expression in Cell Pellets and Secreted Exosomes

MiRNAs are among the most important EV cargoes that develop critical functions on cells and have demonstrated altered levels in many pathologies. Given this crucial role, we analysed the levels of several key miRNAs in podocyte biological functions and DN disease in podocyte cell pellets and in exosomes derived from these cells to observe their alterations when Rabs were silenced and podocytes were subjected to glucose stress.

We found that for *RAB3A* silencing, levels of miR-21-5p in the cell pellet were increased under HG scr (1.94 FC), *p* < 0.05) or HG siRab3A (2.22 FC, *p* < 0.05). Conversely, in secreted exosomes, these levels were augmented under HG treatment and when *RAB3A* was silenced (3.12 FC and 3.42 FC, respectively, *p* < 0.05 for both), but lower in the HG siRab3A group than in NG siRab3A (1.00 FC, *p* < 0.05), suggesting an influence of *RAB3A* silencing under an HG stress condition (Figure 9A,B). For miR-26a-5p, we found the same effect as for miR-21-5p at the cell pellet level, an increase in HG scr group (2.01 FC, *p* < 0.05), but a potent decrease under siRab3A silencing without glucose treatment (−1.56 FC, *p* < 0.05), which increased when silenced Rab was treated with HG (1.04 FC, *p* < 0.05 compared to NG siRab3A). Turning to exosomes, miR-26a-5p expression levels were highly decreased under siRab3A (−22.2 FC, *p* < 0.05), while podocytes with *RAB3A* silenced and subjected to HG showed raised levels compared to NG siRab3A (−3.05 FC, *p* < 0.05) (Figure 9C,D). A drop in miR-31-5p cell pellet levels was found when *RAB3A* was silenced with or without glucose treatment (−2.23 FC, *p* < 0.01 and −1.56 FC, *p* < 0.05 compared to NG scr and HG scr groups, respectively), while exosomal levels were undetectable in most samples (Figure 9E).

In the case of *RAB27A* silencing, exosomal miRNA levels were undetectable in most samples. In the cell pellet, miR-21-5p levels were increased in the HG groups (2.06 FC, *p* < 0.01 and 2.13 FC, *p* < 0.05) (Figure 10A). MiR-26a-5p levels were decreased in the HG scr group compared to NG scr (−3.33 FC, *p* < 0.05), whereas under *RAB27A* silencing, miR-26a-5p levels increased when cells were subjected to HG treatment (1.60 FC, *p* < 0.05) (Figure 10B). Finally, miR-31-5p levels showed an increase under HG treatment (1.99 FC, *p* < 0.05), but in siRab27A groups, both with and without glucose, they fell to levels similar to those of the NG scr group (Figure 10C).

## 4. Discussion

Podocyte damage is central to the development of glomerular injury. As major regulators of the glomerular filtration barrier, intra and intercellular communication between podocytes and other renal cell types is crucial, and EVs are key counterparts in this process. In this context, Rab GTPases play a role in controlling vesicular trafficking and can affect an individual’s health status or contribute to disease. For this reason, we therefore investigated the role of silencing the Rab3A/27A system in glucose-induced podocyte injury to shed some light on the system’s role in glomerular damage under diabetic conditions.

Few previous studies have analysed the Rabs silencing part of this system at the renal level. Armelloni et al. studied the effect of *Rab3A* knockout mice chronically fed with a HG diet, resulting in increased podocyte damage and leading to proteinuria [32]. Conversely, Li et al. showed that inhibiting *Rab27A* caused an improvement in renal tubular epithelial cells by reducing inflammation and diabetic kidney disease development [37]. In accordance with this, our previous work showed the seemingly important role of the Rab3A/27A system in podocyte damage when cells were subjected to stress. In this prior work we found an increase of Rabs under glucose overload, accompanied by a general decrease in podocyte differentiation markers, indicating that a dedifferentiated state could occur under stress situations and that the Rab3A/27A system may participate in this process [29].

Coinciding with our previous results, we have observed that silencing *RAB3A* on podocytes and subjecting them to glucose overload is associated with a general increase in podocyte markers, indicating an absence of a dedifferentiation state in podocytes. Rab3A is involved in the exocytosis of secretory vesicles and granules [33,38], but studies of its function have revealed conflicting roles in exocytosis processes [39]. Some studies indicate a role for Rab3A as a negative regulator of exocytosis [40,41], while in others it has been found to enhance vesicle secretion [42,43], and the occurrence of these opposing functions is reported to depend on the secretory cell type [39]. Our results evidence that through its function, Rab3A could participate in certain mechanisms of damage under diabetic conditions. The effect of *RAB27A* silencing and treatment on podocyte markers is not as marked as for Rab3A, and their expression is not as pronounced as with *RAB3A* silencing. Both Rabs have common and distinct roles in vesicle traffic, but interestingly, some publications report a cooperative role between the two Rabs to achieve vesicle exocytosis at plasma membranes [25,44]. In line with this, our observation is that when one Rab is silenced without glucose treatment, the expression levels of the other Rab and their common effector Rph3A generally decrease, but under glucose overload these expression levels increase, indicating a potential compensatory mechanism of Rabs in a cellular stress situation.

Remarkably, we found common features in apoptosis, the cell structure of podocytes, and vesicular transport markers. Generally, under silenced Rabs and no treatment conditions, we observed an increase in cell death and a decrease in anisotropy, demonstrating F-actin fibre disorganisation. We also found changes in the CD63 pattern, from dispersal in the cytoplasm to accumulation in the perinuclear zone. As previously shown, processes such as apoptosis increase under treatment [29,45], yet when cells are subjected to glucose overload and Rabs are silenced, adverse processes such as apoptosis, cellular structure organisation and CD63 accumulation are apparently reduced. These results suggest that different mechanisms may be involved when Rabs are silenced, depending on whether the cell is under stress or not. Under normal glucose conditions, Rab silencing could decrease the secretion of CD63-positive vesicles, as seen by their pattern of accumulation in the perinuclear zone. Nevertheless, when cells are damaged by treatment, the signals they would be sending could be negative and are partially aborted by silencing the Rab system. In the context of DN, these results demonstrate a promising role of the Rab3A/27A system, which emerges as a potential means to diminish the detrimental processes that occur in podocytes.

Furthermore, we investigated the levels of several miRNAs with involvement in DN and podocyte damage. Involved in many cellular processes, miRNAs are important EV cargoes. In fact, the differential expression levels of miRNAs have been associated with renal damage; they have therefore been used as biomarkers of the disease and/or proposed as potential therapeutic targets [35,46,47,48]. Taking this into account, we sought to determine their levels in cellular pellets and EVs from podocytes silenced and treated with glucose. Our results showed that for both silenced Rabs, miR-21-5p cell pellet levels increased in both HG-treated podocytes, suggesting their involvement in podocyte damage as positive or negative influencers. Interestingly, these levels in exosomes from *RAB3A*-silenced podocytes were decreased in HG siRab3A, pointing to a decrease in this miRNA loading to vesicles when *RAB3A* is silenced and treated, whereas under silencing with no treatment, these levels were greatly increased. MiR-21-5p is one of the most characterised miRNAs and has demonstrated a pathological role in renal physiology, although its role in pathogenesis is controversial [49,50]. It has shown increased levels in DN models, and is associated with microalbuminuria, inflammation, and renal fibrosis [51]. However, in another study, streptozotocin-induced diabetic mice showed increased levels of miR-21, and lack of miR-21 was associated with accelerated glomerular injury and increased proteinuria, with a reduction in podocyte density, which leads to the conclusion that miR-21 ameliorates TGF-β1 and hyperglycaemia-induced glomerular injury through repression of pro-apoptotic signals [52].

Another miRNA related to the TGF-β1 signalingsignalling pathway is miR-26a-5p, which was found at increased levels in the cellular fraction of HG-treated only, and HG and silenced group, in both silenced Rabs. MiR-26a-5p has been proposed to have a protective role in diabetes and DN, and has thus been suggested as a treatment means [35,53,54,55]. In this way, Koga et al. reported that downregulation of miR-26a favours progression to DN by enhancing TGF-β/CTGF signalling [56]. Conversely, miR-26a-5p levels in exosomes were potently downregulated in the HG group that was partially recovered in the HG siRab3A group. These results coincide with those of Li et al., who found decreased levels of this miRNA in exosomes from HG-treated fibroblasts [57]. Additionally, we found similar results in urinary exosomes from hypertensive patients with renal damage and in exosomes from TGF-β1-treated podocytes [35]. Finally, miR-31-5p was increased in the HG group, and its levels decreased significantly with Rab silencing, as well as in treated and silenced podocytes, which suggests that Rab silencing seems to influence miR-31-5p expression in podocytes. Although little evidence has been accumulated to support this Rab system as a regulator of miR-31-5p, one study proposed Rab27A as a target for miR-31-5p, together with target prediction tools such as Targetscan, a line which deserves further investigation [58]. Moreover, this miRNA has been linked with DN and was proposed as a serum disease biomarker related with inflammation [59].

The results of this study should be interpreted within the framework of several limitations. One of these is the presence of transient transfection, which restricts the performance time of experiments to a limited silencing-effect frame time. Another is the lack of animal studies with which to test the findings of the study. Nevertheless, this is an experimental study using a human immortalised cell line that mimics the conditions produced in DN disease. As such, it serves as a point of reference for future experiments centred on the effect of silencing the Rab3A/27A system and its influence on vesicular traffic.

## 5. Conclusions

In summary, our findings show that silencing the Rab3A/27A system in podocytes and subjecting them to HG-induced injury exerts strong effects by affecting the differentiation and apoptosis processes, cytoskeleton organisation, and the distribution of CD63-positive vesicles. Moreover, miRNAs relevant for DN showed altered expression under silencing and treatment with glucose. All these findings point to the Rab3A/27A system as a key participant in podocyte injury and in vesicular traffic regulation in the context of diabetes. Therefore, further investigation of the modulation of the Rab3A/27A system could provide novel targets to ameliorate the podocyte damage produced in DN. Investigating the miRNA EV cargoes may reveal pathways for finding out new injury mechanisms.

## Figures and Tables

**Figure 1 biology-12-00690-f001:**
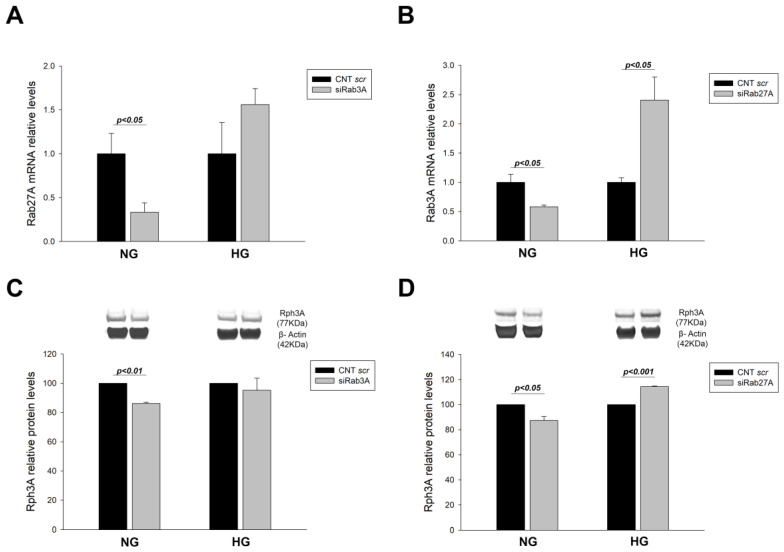
Effect of RAB3A and RAB27A gene silencing and treatment with high glucose on other components of the Rab3A/27A system. (**A**) mRNA levels of Rab27A when Rab3A is silenced under NG and HG conditions. (**B**) mRNA levels of Rab3A when Rab27A is silenced under NG and HG conditions. (**C**,**D**) Protein levels of Rph3A when Rab3A and Rab27A are silenced, respectively. CNT scr: control scramble; NG: normal glucose; HG: high glucose; siRab3A: Rab3A siRNA; siRab27A: Rab27A siRNA; Rph3A: Rabphilin3A. Data are shown as mean ± SEM (standard error of the mean). N = 5 for each group. For the CNT scr group, mRNA levels are normalised to 1 and protein levels to 100.

**Figure 2 biology-12-00690-f002:**
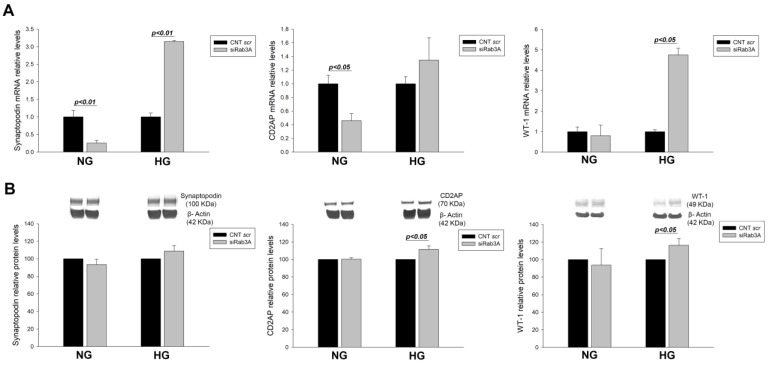
Effect of *RAB3A* gene silencing and treatment with high glucose on podocyte markers. (**A**) mRNA levels of podocyte markers synaptopodin, CD2AP and WT-1 when Rab3A is silenced under NG and HG conditions. (**B**) Protein levels of podocyte markers synaptopodin, CD2AP and WT-1 when Rab3A is silenced under NG and HG conditions. CNT scr: control scramble; NG: normal glucose; HG: high glucose; siRab3A: Rab3A siRNA. Data are shown as mean ± SEM (standard error of the mean). N = 5 for each group. For the CNT scr group mRNA levels are normalised to 1 and protein levels to 100.

**Figure 3 biology-12-00690-f003:**
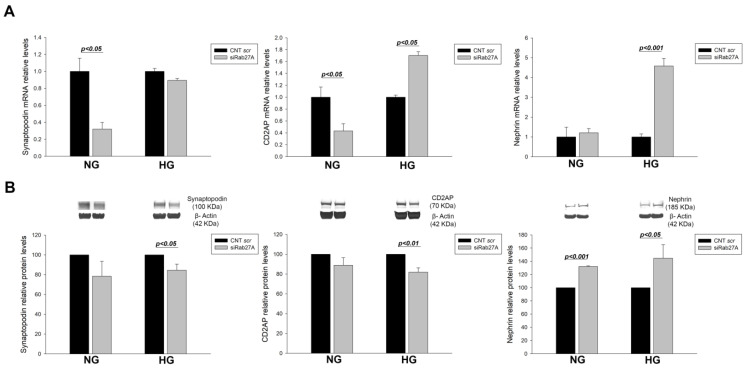
Effect of *RAB27A* gene silencing and treatment with high glucose over podocyte markers. (**A**) mRNA levels of podocyte markers synaptopodin, CD2AP and nephrin when *RAB27A* is silenced under NG and HG conditions. (**B**) Protein levels of podocyte markers synaptopodin, CD2AP, and nephrin when *RAB27A* is silenced under NG and HG conditions. CNT scr. control scramble; NG: normal glucose; HG: high glucose; siRab27A: Rab27A siRNA. Data are shown as mean ± SEM (standard error of the mean). N = 5 for each group. For the CNT scr group, mRNA levels are normalised to 1 and protein levels to 100.

**Figure 4 biology-12-00690-f004:**
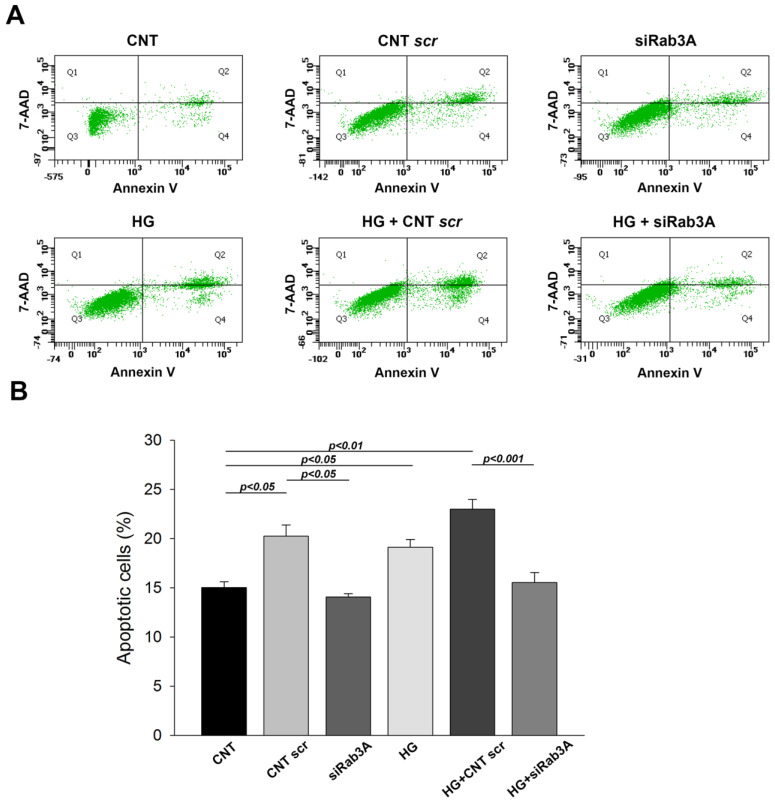
Influence of *RAB3A* gene silencing and treatment with high glucose on podocyte apoptosis. (**A**) Flow cytometry images showing cells double-stained with annexinV/-amino-actinomycin (7-AAD). The cytograms of cells with lost intact cell membranes that bound 7-AAD and excluded annexin V are shown in quadrant 1 (Q1). Cells with advanced stages of apoptosis or necrosis were both annexin V- and 7-AAD-positive and are shown in quadrant 2 (Q2). Viable cells that did not bind annexin V or 7-AAD are depicted in quadrant 3 (Q3). Early apoptotic cells, positive for annexin V but with still intact cell membranes (7-AAD negative), are shown in quadrant 4 (Q4). (**B**) Graphs showing apoptosis percentages. CNT: control; CNT scr: control scramble; NG: normal glucose; HG: high glucose; siRab3A: Rab3A siRNA. Data are shown as mean ± SEM (standard error of the mean). N = 5 for each group.

**Figure 5 biology-12-00690-f005:**
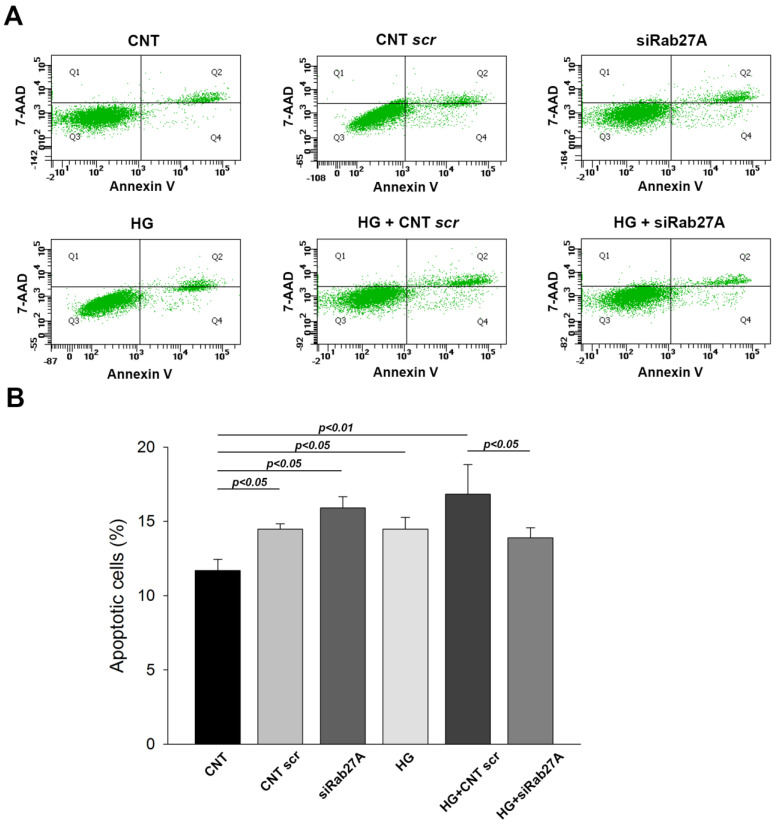
Influence of *RAB27A* gene silencing and treatment with high glucose over podocyte apoptosis. (**A**) Flow cytometry images showing cells double-stained with annexinV/-amino-actinomycin (7-AAD). The cytograms of cells with lost intact cell membranes that bound 7-AAD and excluded annexin V are shown in quadrant 1 (Q1). Cells at advanced apoptosis or necrotic stages were both annexin V- and 7-AAD-positive and are shown in quadrant 2 (Q2). Viable cells that did not bind annexin V or 7-AAD are depicted in quadrant 3 (Q3). Early apoptotic cells, positive for annexin V but with still intact cell membranes (7-AAD negative), are shown in quadrant 4 (Q4). (**B**) Graphs showing apoptosis percentages. CNT: control; CNT scr: control scramble; NG: normal glucose; HG: high glucose; siRab27A: Rab27A siRNA. Data are shown as mean ± SEM (standard error of the mean). N = 5 for each group.

**Figure 6 biology-12-00690-f006:**
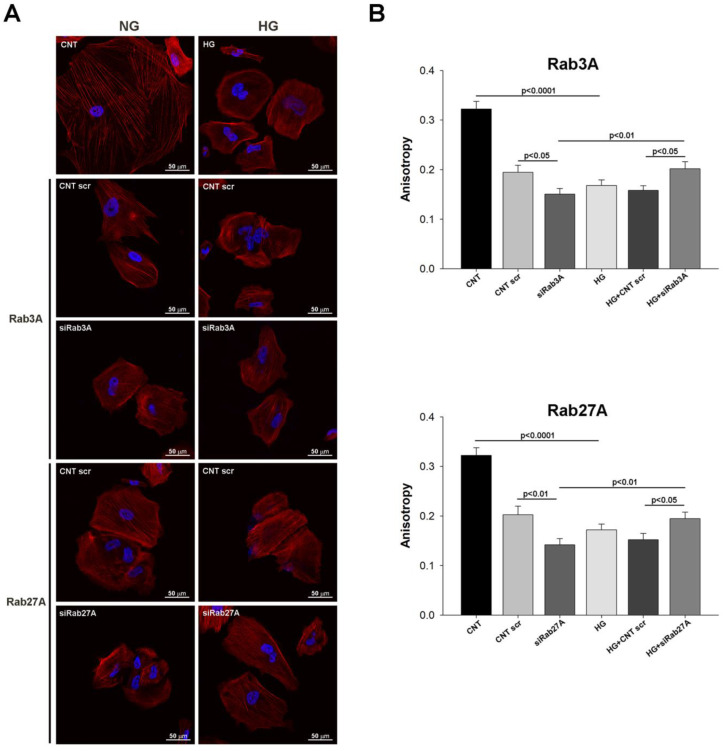
Impact of Rab3A/27A system gene silencing and treatment with high glucose on the podocyte structure. (**A**) Confocal images of podocytes subjected to the gene silencing of *RAB3A* and *RAB27A* and under glucose overload, showing F-actin fibre staining with phalloidin. (**B**) Graphs showing the anisotropy index in each group for *RAB3A* and *RAB27A* silencing. CNT: control; CNT scr: control scramble; NG: normal glucose; HG: high glucose; siRab3A: Rab3A siRNA; siRab27A: Rab27A siRNA. Data are shown as mean ± SEM (standard error of the mean). N = 5 for each group.

**Figure 7 biology-12-00690-f007:**
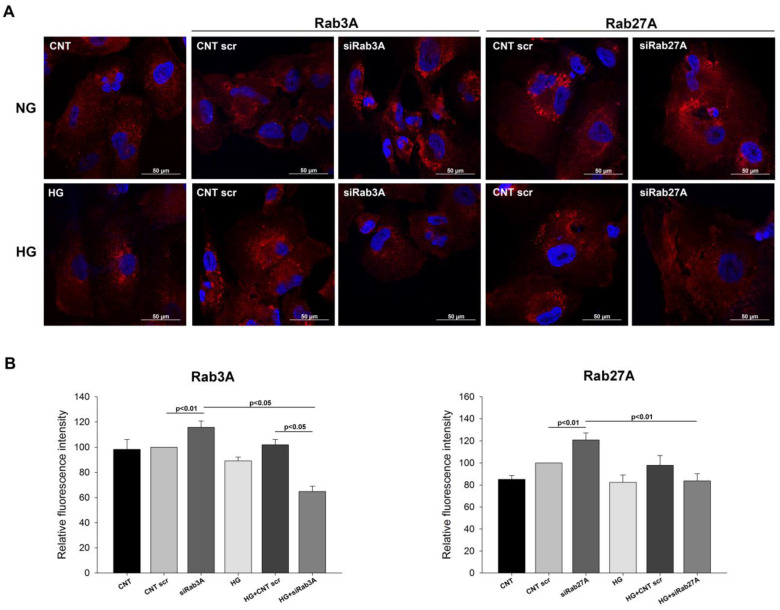
Analysis of CD63 intensity and distribution in silenced and glucose overload-treated podocytes. (**A**) Confocal images of podocytes treated with siRab3A and siRab27A and under glucose overload, showing CD63 staining and cell distribution. (**B**) Graphs showing the fluorescence intensity of CD63 in each group for Rab3A and Rab27A silencing. CNT: control; CNT scr: control scramble; NG: normal glucose; HG: high glucose; siRab3A: Rab3A siRNA siRab27A: Rab27A siRNA. Data are shown as mean ± SEM (standard error of the mean). N = 5 for each group.

**Figure 8 biology-12-00690-f008:**
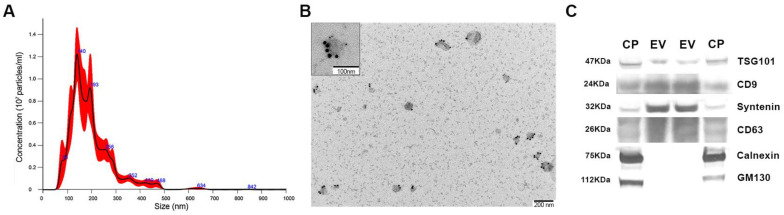
Characterization of podocyte cell culture medium EVs. (**A**) Graph showing the NTA results with the concentration and size of EV particles. (**B**) TEM micrographs of CD9 (6 nm) and CD63 (15 nm) with immunogold labelling. (**C**) Western blot analysis of EV markers (TSG101; CD9, syntenin, CD63) and cell organelle markers (calnexin and GM130) in EV and CP. CNT: control; CNT scr: control scramble; CP; cell pellet; EV: extracellular vesicles; GM130: Golgi matrix protein 130; NG: normal glucose; NTA: nanoparticle tracking analysis; HG: high glucose; siRab3A: Rab3A siRNA; siRab27A: Rab27A siRNA; TEM: transmission electron microscopy; TSG101: tumour susceptibility gene 101 protein. N = 5 for each group.

**Figure 9 biology-12-00690-f009:**
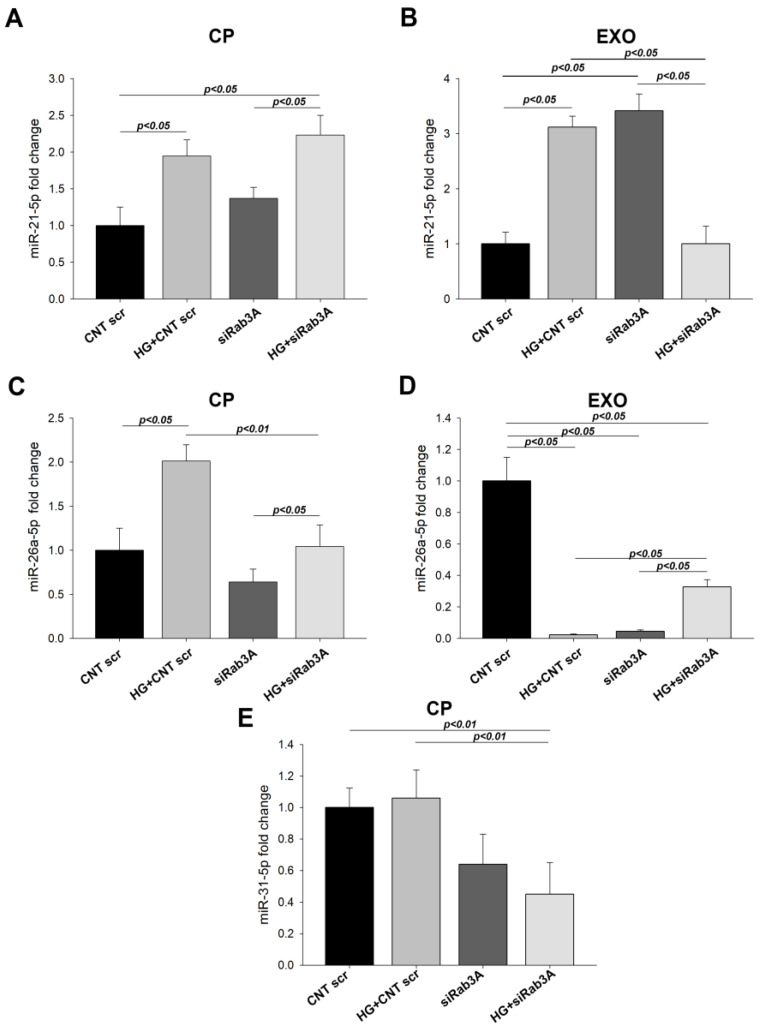
Analysis of miRNA levels in *RAB3A* silenced and glucose overload-treated podocytes and EVs. (**A**) miR-21-5p CP levels. (**B**) miR-21-5p EXO levels. (**C**) miR-26a-5p CP levels. (**D**) miR-26a-5p EXO levels. (**E**) miR-31-5p CP levels. CNT: control; CNT scr: control scramble; CP: cell pellet; EXO: exosome; HG: high glucose; siRab3A: Rab3A siRNA. Data are shown as mean ± SEM (standard error of the mean). N = 5 for each group. miRNA levels for CNT scr group are normalised to 1.

**Figure 10 biology-12-00690-f010:**
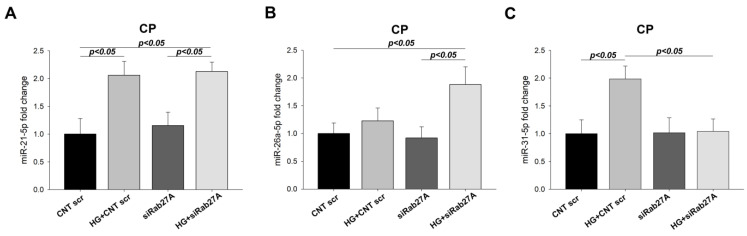
Analysis of miRNA levels in *RAB27A* silenced and glucose overload-treated podocytes. (**A**) miR-21-5p CP levels. (**B**) miR-26a-5p CP levels (**C**) miR-31-5p CP levels. CNT: control; CNT scr: control scramble; CP: cell pellet; HG: high glucose; siRab27A: Rab27A siRNA. Data are shown as mean ± SEM (standard error of the mean). N = 5 for each group. miRNA levels for CNT scr group are normalised to 1.

**Table 1 biology-12-00690-t001:** Primer sequences for RT-qPCR.

Target (Gene Name)	Size (bp)	Sequence 5′ → 3′
*ACTB*	20	F TGGAGAAAATCTGGCACCAC
	22	R CATGATCTGGGTCATCTTCTCG
*B2MG*	21	F TCCAGCGTACTCCAAAGATTC
	21	R GTCAACTTCAATGTCGGATGG
*CD2AP*	20	F AGGCATGGGAATGTAGCAAG
	21	R TGACGCTTCTTGGTCTTCTTC
*NPHS1*	20	F TGCAGTTTCCCCCAACTAAC
	19	R ACGCTGACGCATGTCAAGT
*RAB3A*	23	F CCTCATGTATGACATCACCAACG
	22	R CCTCAAAGAACTCGAACCCAAG
*RAB27A*	24	F AAAGAGTGGTGTACAGAGCCAGTG
	24	R GTCGTTAAGCTACGAAACCTCTCC
*SYNPO*	20	F GCCGCAAATCCATGTTTACT
	20	R CTCATCCGCTGTCTGTACCA
*WT-1*	20	F TTCGCAATCAGGGTTACAGC
	20	R AATGAGTGGTTGGGGAACTG

*ACTB*: actin β; *B2MG*: β-2-microglobulin; *CD2AP*: CD2-associated protein; *NPHS1*: nephrin; *RAB3A*: Ras-related protein Rab-3A; *RAB27A*: Ras-related protein Rab-27A; *SYNPO*: synaptopodin; *WT-1*: Wilms tumor-1.

## Data Availability

All the data supporting the reported results are contained within this article and its Appendix A.

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
