# Peer review of "Rab3A/Rab27A System Silencing Ameliorates High Glucose-Induced Injury in Podocytes"

_biology, 2023, doi:10.3390/biology12050690_

Round 1

Reviewer 1 Report

This paper is an interesting study that describes changes in podocytes after manipulation of the vesicular trafficking protein system Rab3A/27A. The results of this work will be useful to understanding the mechanisms of podocyte dysfunction development in diabetic nephropathy, which is of extreme importance given the increasing incidence of diabetes mellitus worldwide. Despite the absence of in vivo experiments, the article has an important scientific significance for researchers working in a similar direction.

Unfortunately, there are no morphological data on the structure of podocytes revealed by transmission electron microscopy in the work: it would be interesting to trace the changes in intracellular vesicles (such as early and late endosomes) under conditions of RAB3A and RAB27A gene silencing. However, this is not a drawback of the work, but can serve as an additional direction in planning further experiments.

Finally, it is necessary to correct Figure 8B: the scale bars on TEM images are difficult to distinguish, they should be increased or the scale value should be additionally written in the figure caption.

Author Response

First, we want to thank Reviewer 1 for considering our manuscript interesting and useful for uncovering the mechanisms of podocyte damage in diabetic nephropathy. We agree with Reviewer 1 that we have not morphological data of podocyte structure when Rab3A/Rab27A system is silenced. It would have been interesting to trace the vesicular traffic changes by labelling the early and late endosome pathways under silencing conditions. So, it is an interesting experiment that we will take into account for our future studies deepening into the involvement of Rab3A/Rab27A system in podocyte dysfunction.

Regarding the scale bars on TEM images from the Figure 8B, we have modified them and the numbers are much bigger, we think that now are more clearly shown.

Finally, we want to thank Reviewer 1 for their suggestions and comments regarding our manuscript that have helped to improve it.

Reviewer 2 Report

Dear Authors,

Manuscript ID biology-2344284

Diabetic nephropathy is one of the leading causes of end-stage kidney disease and develops huge healthcare burdens worldwide.

Here are my suggestions:

1. Authors can add a few lines on:

What causes damage to podocytes? in the introduction

2. Give the expansion of uncommon abbreviations at the initial usage in your manuscript.

3. Weave your sentences coherently to explain the objective of your research clearly.

4. There were many incomplete sentences and unnecessary repetition of words. Please check them.

5. Take care of the usage of scientific units in methodologies.

6. Language correction is essential for this manuscript.

7. Title needs a correction. Recheck.

8. Take care of the formatting errors.

9. Verify the whole manuscript for errors.

10. Focus on results, discussion and conclusion. Needs clarity.

Author Response

RESPONSE TO REVIEWER 2

Diabetic nephropathy is one of the leading causes of end-stage kidney disease and develops huge healthcare burdens worldwide.

First, we want to thank Reviewer 2 for your revision that helps to improve the quality of the manuscript.

Here are my suggestions:

  1. Authors can add a few lines on:

What causes damage to podocytes? in the introduction

In accordance with Reviewer 2 suggestion, we have added some sentences and references in the introduction regarding the causes of podocyte damage in diabetic nephropathy.

Introduction section, pages 2-3:

“Physiologically, renal function is compromised, in part, by the weakening of glomerular filtration barrier [4]. Podocytes are highly specialized cells that, in healthy kidneys, create an interdigitated network through foot processes surrounding the basal membrane to prevent plasma proteins from entering the urinary ultrafiltrate [5]. Permanent hyperglycaemia stimulates diverse signalling pathways and mechanical insults that act in an additive manner, thereby accelerating the podocyte damage [6,7]. Different molecular pathways lead to podocyte dysfunction through the inhibition of autophagy, accelerating epithelial to mesenchymal transition (EMT) or podocyte hypertrophy [8-11]. Sirt 1 signalling pathway is closely related to damage in DN. Under increased glucose levels the expression of Sirt1 is inhibited, which increases oxidative stress and triggers apoptosis [12,13]. Moreover, in a hyperglycaemia environment, many factors activate transforming growth factor- (TGF-) β1 which activates downstream pathways that ultimately stimulates the secretion of VEGF increasing of glomerular permeability [14]. In addition, VEGF, reactive oxygen species, and AngII can stimulate TGF-β, leading to accumulation of mesangial extracellular matrix, podocyte hypertrophy and changes on its morphology [15,16]. When damaged, podocytes suffer foot process effacement, apoptosis and detachment from the basal membrane, negatively affecting the filtration process [17,18].”

  1. Give the expansion of uncommon abbreviations at the initial usage in your manuscript.

We have revised the abbreviations used in the manuscript and we have explained the uncommon ones at the first usage.

  1. Weave your sentences coherently to explain the objective of your research clearly.

We have better explained the objective of our study:

“In this context, we aimed to silence gene expression of the Rab3A/27A system in human immortalized podocytes stressed by high glucose, to gain more in-depth insight into their effect on podocyte morphology, apoptosis and function and ascertain how this silencing could affect vesicular traffic and their cargos.”

  1. There were many incomplete sentences and unnecessary repetition of words. Please check them.

Taking into account these recommendations, we have modified some sentences and tried to better explain them.

  1. Take care of the usage of scientific units in methodologies.

We have revised and corrected the SI units in the methods section.

  1. Language correction is essential for this manuscript.

As the reviewer rightly indicates, the text needed extensive language professional editing and we have done so, undergoing substantial changes. We have applied all the corrections proposed by the English editing service to improve the writing and comprehension of the manuscript (the main changes have been indicated by track changes).

  1. Title needs a correction. Recheck.

We have modified the title as follows:

Rab3A/Rab27A system silencing ameliorates injury in glucose overload podocytes

“Rab3A/Rab27A system silencing ameliorates high glucose-induced injury in podocytes”.

  1. Take care of the formatting errors.

As suggested, we have revised the formatting errors and corrected them. The changes are highlighted with control changes along the manuscript.

  1. Verify the whole manuscript for errors.

We have verified all the manuscript and corrected the errors. Changes are highlighted with track changes too.

  1. Focus on results, discussion and conclusion. Needs clarity.
    We have clarified some sentences from these parts for better understanding.

Finally, we want to apologize for the grammatical, language and formatting errors found in the manuscript and we want to thank Reviewer 2 for their pertinent comments and suggestions that have helped to better understand the manuscript and their results.

Reviewer 3 Report

The paper was well written. The methodology was sufficient. The RAB system has various components but the reviewer left wondered why the authors wants to understand the role of RAB3A and RAB27A. The data presented was hard to read. The presentation of the data needs to improved by showing the siRNA knockdown separated and glucose stimulation separated.  It is quite interesting that complementary relation between RAB3A and RAB27A.

The system although seems important to the podocyte but not required for high glucose mediated changes. Because the effect seems very mild not even moderate. Authors need to try another strong cell death model such as TGFb or growth hormone. They both induce cell death in podocytes and cause almost similar phenotype as high glucose does, although not compared by this group (doi;10.1016/j.bbamcr.2022.119391,doi;10.1096/fj.202200923R, 10.1038/s41419-021-03643-6). The suggested studies used proteomics and transcriptomics to observe whether the the factors change RAB system in podocytes. 

It would be interesting if the authors can comment on how the RAB system is finally involved in regulating podocyte death with additional evidence.

Author Response

RESPONSE TO REVIEWER 3

The paper was well written. The methodology was sufficient. The RAB system has various components but the reviewer left wondered why the authors wants to understand the role of RAB3A and RAB27A. The data presented was hard to read. The presentation of the data needs to improved by showing the siRNA knockdown separated and glucose stimulation separated. It is quite interesting that complementary relation between RAB3A and RAB27A.

The system although seems important to the podocyte but not required for high glucose mediated changes. Because the effect seems very mild not even moderate. Authors need to try another strong cell death model such as TGFb or growth hormone. They both induce cell death in podocytes and cause almost similar phenotype as high glucose does, although not compared by this group (doi;10.1016/j.bbamcr.2022.119391,doi;10.1096/fj.202200923R, 10.1038/s41419-021-03643-6). The suggested studies used proteomics and transcriptomics to observe whether the the factors change RAB system in podocytes.

It would be interesting if the authors can comment on how the RAB system is finally involved in regulating podocyte death with additional evidence.

First of all we want to thank Reviewer 3 for considering our manuscript well written and the methods appropriate. Regarding the question about why we decided to study this Rab3A/Rab27A system, we cite in the introduction and discussion several studies deciphering the functions of this system in cellular communication and the previous findings of our group in podocyte injury. So, we think that was important to deepen in Rab3A/Rab27A system study by modifying its gene expression.

We understand that we have many experimental conditions and the manuscript is hard to read. In all these conditions we wanted to show the effect of the silencing (when cells are not subjected to high glucose) and the effect of this silencing and glucose altogether. We have previously shown the effect of the glucose itself (Martinez-Arroyo O et al 2020 AJP-Renal Physiol), so we did not want to complicate the results with more comparisons and in addition, this was not the focus of the study. According to the results section, we have modified some sentences to better explain and clarify them. These changes are highlighted with control changes.

We are conscious that treatment with TFGb and growth hormone have important impacts on cell apoptosis, and the studies mentioned by Reviewer 3 are very interesting and decipher important implications of TNFa pathway in podocyte damage. However, in this study we analyzed the effects of the silencing and treatment with glucose over some functions of podocytes, apoptosis among others, not being the main focus of the article. In further studies creating this model with GH stimulation will be interesting to deepen in the effect of Rab3A/Rab27A silencing on apoptosis process. Nevertheless, we have considered important to cite some of the studies proposed by the Reviewer 3 and we have done so in the introduction:

Introduction, pages 2-3:

Physiologically, renal function is compromised, in part, by the weakening of glomerular filtration barrier {Thomas, 2021 #4}. Podocytes are highly specialized cells that, in healthy kidneys, create an interdigitated network through foot processes surrounding the basal membrane to prevent plasma proteins from entering the urinary ultrafiltrate {Reiser, 2016 #5}. Permanent hyperglycaemia stimulates diverse signalling pathways and mechanical insults that act in an additive manner, thereby accelerating the podocyte damage {Lewko, 2009 #53;Podgorski, 2019 #54}. Different molecular pathways lead to podocyte dysfunction through the inhibition of autophagy, accelerating epithelial to mesenchymal transition (EMT) or podocyte hypertrophy {Zhang, 2020 #55;Kaushal, 2019 #56;Wu, 2017 #57;Das, 2018 #58}. Sirtuin 1 signalling pathway is closely related to damage in DN. Under increased glucose levels the expression of Sirtuin 1 is inhibited, which increases oxidative stress and triggers apoptosis {Bible, 2013 #59;Martinez-Arroyo, 2020 #22}. Moreover, in a hyperglycaemia environment, many factors activate transforming growth factor- (TGF-) β1 which activates downstream pathways that ultimately stimulates the secretion of vascular endothelial growth factor (VEGF) increasing of glomerular permeability {Miaomiao, 2016 #61}. In addition, VEGF, growth hormone, reactive oxygen species, and Angiotensin II can stimulate TGF-β, leading to accumulation of mesangial extracellular matrix, podocyte hypertrophy, mitotic effects and changes on its morphology {Susztak, 2006 #62;Chen, 2005 #63} {Nishad, 2021 #64} {Mukhi, 2023 #65}. When damaged, podocytes suffer foot process effacement, apoptosis and detachment from the basal membrane, negatively affecting the filtration process {Assady, 2017 #6;Yin, 2021 #45}.

Finally, and once again, we are very grateful to Reviewer 3 for their comments that we think have improved this manuscript and helped us with the suggestions for applying them in our future studies on Rab3A/Rab27A system.

Reviewer 4 Report

esultados de traducción

Resultado de traducción

      Olga Martinez-Arroyo's article is interesting, well designed and presented some doubts and suggestions. 1. It is the first time that I observe Graphical Abstract in MDPI journals. It's right? 2. Can you indicate which was the research committee that authorized the project? What registration number do you have? I observe that they have a lot of financing, but I do not observe the authorization of a research committee. 3. The western blot results, in Figure 1C and 1D, were performed in triplicate? Can you show western blot results in triplicate? 4. The western blot results, in Figure 2B, were performed in triplicate? Can you show western blot results in triplicate? 5 Western blot results, in Figure 3B, were performed in triplicate? can you show the western blot results in triplicate? 6. Immuhistomy images are well done 7. In figure 8C, which is the control?

Author Response

RESPONSE TO REVIEWER 4

Olga Martinez-Arroyo's article is interesting, well designed and presented some doubts and suggestions.

First of all, we want to thank Reviewer 4 for considering our work interesting and well designed.

  1. It is the first time that I observe Graphical Abstract in MDPI journals. It's right?

As a new item in Biology-basel, the journal encourages to submit a graphical abstract that represents a synthesis of the main findings of the study without being a part of a Figure. So we decided to elaborate a graphical abstract to show more clearly our findings.

  1. Can you indicate which was the research committee that authorized the project? What registration number do you have? I observe that they have a lot of financing, but I do not observe the authorization of a research committee.

We sorry for not indicating the Research Committee that approved the study. This research has been conducted by the Ethics Committee from the Hospital Clinico de Valencia with the registration number 2019/063. In accordance with this information, we have included an Institutional Review Board Statement:

Institutional Review Board Statement:  Cell line AB8/13 of conditionally human immortalized podocytes was kindly provided by Prof Moin Saleem of the Children’s Renal Unit and Academic Renal Unit, University of Bristol, Southmead Hospital, Bristol, UK. And written informed consent from the cell donor was previously obtained. This project was approved by the Ethics Committee from the Hospital Clinico de Valencia with the registration number 2019/063.

  1. The western blot results, in Figure 1C and 1D, were performed in triplicate? Can you show western blot results in triplicate? 4. The western blot results, in Figure 2B, were performed in triplicate? Can you show western blot results in triplicate? 5 Western blot results, in Figure 3B, were performed in triplicate? can you show the western blot results in triplicate?

For the western blot experiments, we used a total of 5 independent cell passages for each condition (n=5), as we state in the Figure legends. Apart from this, we run some of the samples twice or three times in different gels and we after quantified their intensities and performed the average of them for the analysis.

  1. Immuhistochemistry images are well done

Thank you for considering these experiments well conducted.

  1. In figure 8C, which is the control?

The control for these western blot experiments is the Cell Pellet (CP), which is the experimental condition used for comparing the intensity of the exosome samples.

Finally, we would like to thank Reviewer 4 for the comments and suggestions that have helped to improve some important ethical and experimental parts of the manuscript.

Round 2

Reviewer 2 Report

Dear Authors,

You have revised this manuscript as per the previous suggestions.

Line no 19:  Mention either system or gene

Despite the corrections, this paper still needs language editing, again check for errors.

Author Response

RESPONSE TO REVIEWER 2

Dear Authors,

You have revised this manuscript as per the previous suggestions.

Line no 19:  Mention either system or gene

Despite the corrections, this paper still needs language editing, again check for errors.

Once again we want to thank Reviewer 2 for his/her comments and suggestions.

According to the suggestion, we have modified the line 19 and put only system in the sentence:

“We have observed that silencing the Rab3A/Rab27A system gene in podocytes experiencing glucose overload has a differential impact on differentiation, cytoskeleton organization, apoptosis, CD63 distribution, and miRNA levels.”

Regarding the English language edition, we have revised it again. The manuscript has been revised by a native person and we hope it is spelled correctly.

All changes are highlighted by Track Changes.

Finally, we are grateful to Reviewer 2 for his/her suggestions that have helped to improve the manuscript.
